

# Beyond graft function impairment after liver transplantation: the prolonged cold ischemia time impact on recurrence of hepatocellular carcinoma after liver transplantation—a single-center retrospective study

Jia Yu[1,2,3,4], Tang Yunhua[1,2,3], Yiwen Guo[1,2,3], Yuqi Dong[1,2,3], Jin long Gong[5], Tielong Wang[1,2,3], Zhitao Chen[1,2,3], Maogen Chen[1,2,3], Weiqiang Ju[1,2,3] and Xiaoshun He[1,2,3]

[1] First Affiliated Hospital, Sun Yat-sen University, Guangzhou, China
[2] Guangdong Provincial Key Laboratory of Organ Donation and Transplant Immunology, Guangzhou, China
[3] Guangdong Provincial International Cooperation Base of Science & Technology (Organ Transplantation), Guangzhou, China
[4] The First Affiliated Hospital of University of South China, Hengyang, China
[5] Hunan Provincial People's Hospital, Changsha, China

Corresponding author
Xiaoshun He, gdtrc@163.com

## ABSTRACT

**Purpose.** Hepatocellular carcinoma (HCC) is one of the malignant tumors responsible for high mortality and recurrence rates. Although liver transplantation (LT) is an effective treatment option for HCC, ischemia-reperfusion injury (IRI) is a contributor to HCC recurrence after LT. Moreover, prolonged cold ischemia time (CIT) is a risk factor for IRI during LT, and there is insufficient clinical evidence regarding the impact of CIT on HCC recurrence after LT.

**Patients and Methods.** This retrospective study analyzed 420 patients who underwent LT for HCC between February 2015 and November 2020 at The First Affiliated Hospital, Sun Yat-sen University. The duration of CIT was defined as the time from clamping of the donor aorta until portal reperfusion.

**Results.** A total of 133 patients (31.7%) experienced tumor recurrence after LT, and CIT > 568 min was the independent risk factor for HCC recurrence (OR, 2.406; 95% CI [1.371–4.220]; $p = 0.002$). Multivariate Cox's regression analysis revealed that the recipients' gender, exceeding Milan criteria, poor differentiation, and alpha-fetoprotein (AFP) ≥400 ng/ml in CIT > 568 min group were independent risk factors for disease-free survival. The peak 7-day postoperative alanine aminotransferase (ALT) level ($p < 0.001$), the peak 7-day postoperative aspartate aminotransferase (AST) level ($p < 0.001$), the peak 7-day postoperative peak total bilirubin (TBIL) level ($p = 0.012$), and the incidence of early allograft dysfunction (EAD) ($p = 0.006$) were significantly higher in the CIT > 568 min group compared to the CIT ≤ 568 min group. Moreover, the amount of fresh frozen plasma (FFP) infusion during the operation increased ($p = 0.02$), and the time of mechanical ventilation postoperative was longer ($p = 0.045$).

**Conclusion**. An effective strategy to improve the prognosis is to reduce CIT; this strategy lowers the recurrence of HCC in patients undergoing LT, especially those within the Milan criteria.

# INTRODUCTION

Hepatocellular carcinoma (HCC) is an aggressive tumor and the fourth leading cause of cancer-related death globally (*Villanueva, 2019*). More than 700,000 people die from HCC yearly. Theoretically, liver transplantation (LT) is an optimum treatment for HCC because it can eliminate both the tumor and underlying liver disease that progresses to cirrhosis. LT can eliminate the primary lesions, such as minute lesions that may be difficult to identify. It can also exterminate the possible cancerization of the rest of the liver. However, early HCC recurrence occurs after LT, which might result from micrometastasis when tumor cells are transported through the circulation (*Choi et al., 2015*). The 10-year recurrence-free survival rate after LT is around 50%–70% (*Adam et al., 2018*; *Pinna et al., 2018*). Prolonged waiting time is common among HCC patients diagnosed with LT, and this issue is a cause of tumor progression (*Mehta et al., 2017*).

The proportion of clinical use of extended criteria donors (ECDs) and the adoption of grafts with prolonged cold ischemia time (CIT) are increasing, considering the shortage of liver grafts, which may consequently lead to serious ischemia-reperfusion injury (IRI). Previous studies have demonstrated that prolonged cold ischemic time can greatly impair graft survival (*Chini et al., 2021*; *Cassuto et al., 2008*; *Busuttil et al., 2005*). Moreover, IRI has a positive correlation with HCC recurrence (*Nagai et al., 2015*; *Grat et al., 2018*). The use of ECD-derived grafts in patients with HCC can also accelerate tumor recurrence after LT (*Chini et al., 2021*; *Cassuto et al., 2008*; *Busuttil et al., 2005*). This retrospective study examined LT in patients with HCC to confirm the relationship between CIT and HCC recurrence.

# PATIENTS AND METHODS

## Study population

This was a retrospective observational study of 508 patients with HCC (diagnosed by radiographic imaging and/or alpha-fetoprotein (AFP) levels, who presented for LT. From February 2015 to November 2020, the patients were admitted to the Department of Organ Transplantation; the First Affiliated Hospital, Sun Yat-sen University. Of 508 patients admitted, 88 patients were excluded because of the following issues: death within 30 days after the operation, nonprimary HCC confirmed by pathology, re-transplantation, pediatric liver transplant, and ischemic-free LT. All liver grafts were procured from donors after brain death or cardiac death. Informed consent to participate in the research was obtained. This study was approved by the Ethics Committee of the First Affiliated Hospital

of Sun Yat-sen University (2022476), and was conducted in accordance with the principles of the Declaration of Helsinki. All the patients had signed human participant information consent. The duration of CIT was defined as the time from clamping of the donor aorta until portal reperfusion. Warm ischemia time (WIT) was calculated from the time when systolic blood pressure was <60 mm Hg and upon lying down (*Kalisvaart et al., 2021*). Immunosuppression was maintained with tacrolimus and mycophenolate mofetil. The recurrence of HCC was diagnosed by the abdomen and chest-computed tomography or magnetic resonance imaging and by AFP levels.

The recurrence time was calculated from the date of LT to the recurrence identification. Survival time was calculated from the date of LT to the death of the last follow-up period. Clinical characteristics of the donors included age, gender, donations after cardiac death (DCD), and laboratory data. Clinical characteristics of recipients included age, gender, comorbidity, indication for LT, laboratory data, model for end-stage liver disease (MELD) scores, treatment before LT contained neoadjuvant treatment, transarterial chemoembolization, radiofrequency ablation, and hepatectomy. Tumor information included pretransplant AFP level, tumor size, number of the tumors, microscopic vascular invasion, and tumor grade. Operative factors comprised operative time, volume of packed red blood cells (PRBC) transfusion, intraoperative blood loss, duration of anhepatic phase, CIT, and WIT. Post-operative factors included early allograft dysfunction (EAD), biliary complications, and acute rejection. The EAD is defined as the bilirubin level $\geq$10 mg/dL on day 7, international normalized ratio $\geq$1.6 on day 7, and alanine aminotransferase (ALT) or aspartate aminotransferase (AST) level >2000 IU/L within the first 7 days after LT, as proposed by *Olthoff et al. (2010)*. According to clinical relevance, AFP threshold levelwere set to 400 ng/ml.

## Statistical analysis

Continuous variables that followed the normal distribution were expressed as means $\pm$ standard deviations and were compared using independent-samples $t$-tests. Moreover, median and interquartile ranges were compared using the Mann–Whitney $U$ test for the 2-group. The Chi-square test was used to compare categorical variables given as numbers and percentages. Receiver operating characteristic (ROC) curves were plotted to determine the optimal cutoffs of continuous factors in predicting recurrence. A logistic regression model was used for multivariate analysis. The recurrence-free survival rate was calculated from the date of LT to the date of tumor recurrence. The data were right-censored if no evidence was found to confirm tumor recurrence on the day of the last follow-up. Multivariate analysis was used to identify significant variables in the univariate analysis of recurrence. All analyses were performed using SPSS software for Windows (version 26.0, SPSS Inc, IBM Corp, Chicago, IL, USA), and the level of significance was set at $p < 0.05$.

# RESULTS

## Overview of demographic characteristics

A total of 420 LT cases that met the inclusion criteria were analyzed (Table 1). The donors were 315 male (75.0%), and their mean age was $36.98 \pm 14.35$ years. The majority of donors originate from donation after brain death donors (DBD), while donation after cardiac death (DCD) donors account for 7.1% of the total with a count of 30. Median levels of AST, ALT, and TBIL before harvest were 59 U/L (interquartile range: 30–106.99), 52.5 U/L (interquartile range: 27–95.5), 16.8 umol/L (interquartile range: 11.23–29.00), respectively. The median CIT was 418.5 min, and the median WIT was 5 min. The median age of the recipients was 52.3 (range of 21–75), and more than half of the recipients were male (52.1%, $n = 219$). The median of the last AFP level before LT was 29.17 ng/ml (interquartile range: 5.26–541.65). In addition, 281 recipients (66.9%) exceeded the Milan criteria. The mean size of the tumor was 53.21 mm (SD $\pm$ 40.23), and the majority of patients (84.3%) had been diagnosed with the hepatitis B virus. The mean of the MELD score was 14.57 (SD $\pm$ 40.23). Median AST, ALT, and TBIL peak levels in 7-day postoperative were 435 U/L (interquartile range: 232.75–993.25), 653 U/L (interquartile range: 345.25–1,023.5), and 105.45 umol/L (interquartile range: 60.235–174.075), respectively. The incidence of EAD was 151(36.0%).

## Effects of CIT on IRI

We grouped the patients into two: CIT > 568 min and CIT $\leq$ 568 min according to the optimal cutoff value of the ROC, peak ALT/AST levels, and total bilirubin on POD7. The differences in the rate of EAD was statistically significant (Table 2). Patients with CIT > 568 min showed significantly higher AST, ALT, and TB levels after LT (AST, 712.5 *vs.* 390 U/L, $p < 0.001$; ALT, 957 *vs.* 593 U/L, $p < 0.001$; TB, 134.05 *vs.* 99.5 umol/L, $p = 0.012$; Figs. 1A–1C). However, no statistical differences were found in Crea level and INR level between the two groups ($p = 0.373$, $p = 0.277$, respectively, Figs. 1D, 1E). The EAD incidence in the group with CIT > 568 min was 50%, which was significantly higher than in the group with CIT $\leq$ 568 min (33.0%) ($p = 0.006$; Fig. 1F). The incidence of biliary complications was 5.6% in the CIT > 568 min group, and 8.6% in CIT $\leq$ 568 min group. The statistical results also demonstrated that prolonged CIT resulted in more fresh frozen plasma (FFP) infusion during the operation ($p = 0.02$) and longer postoperative respiratory support time ($p = 0.045$).

## Risk factors for HCC recurrence

HCC recurrence was observed in 133 patients (31.7%), with the median time to recurrence was 16 months (interquartile range: 9 months-32 months). We compared donor and recipient factors between patients with HCC recurrence ($n = 133$) and those without ($n = 287$). Our findings revealed that in the recurrent group, the proportion of patients with CIT > 568 min significantly increased (26.32% *vs.* 12.89%, $p = 0.001$) (Table 3). Additionally, donors in the recurrence group had significantly higher AST levels before harvest (64 *vs.* 60 U/L, $p = 0.007$), and recipients were younger ($50.82 \pm 10.93$ *vs.* $52.95 \pm 9.8$ years, $p = 0.046$). There was also a significantly higher rate of male donors in

**Table 1** Overview of patients included in the study ($n = 420$).

|  | Number (%) or median (IQR) |
|---|---|
| Donor characteristics |  |
| Age (years) | $36.98 \pm 14.35$ |
| Body mass index (kg/m$^2$) | $22.95 \pm 18.30$ |
| Gender (male) | 315 (75.0%) |
| Warm ischemia time(minutes)(80) | 5 (5–8) |
| Cold ischemia time (min) | 418.5 (351.5–522.25) |
| Sodium level before harvest(mmol/L) | $153.77 \pm 69.98$ |
| ALT level before harvest(U/L) | 52.5 (27–95.5) |
| AST level before harvest(U/L) | 59 (30–106.99) |
| TBIL level before harvest(umol/L) | 16.8 (11.23–29.00) |
| DCD | 30 (7.1%) |
| Recipient characteristics |  |
| Age (years) | $52.28 \pm 10.21$ |
| Body mass index (kg/m$^2$) | $23.27 \pm 3.31$ |
| Gender (male) | 367 (87.4%) |
| Complications |  |
| Hypertension | 56 (13.3%) |
| Diabetes | 54 (12.9%) |
| Coronary heart disease | 9 (2.1%) |
| MELD | $14.57 \pm 8.73$ |
| Child–Pugh classification |  |
| A | 155 (36.90%) |
| B | 178 (42.38%) |
| C | 87 (20.71%) |
| Hepatitis B virus infected | 354 (84.3%) |
| ALT level before LT (U/L) | 38 (23.25–68.75) |
| AST level before LT (U/L) | 50 (35–114) |
| Operation time (min) | $460.34 \pm 107.69$ |
| Intraoperative blood loss (ml) | 1,450 (800–2,000) |
| Intraoperative PRBC transfusions (unit) | 4 (2–7) |
| anhepatic period (min) | $57.21 \pm 30.30$ |
| Peak 7-day postoperative ALT level (U/L) | 653 (345.25–1,023.5) |
| Peak 7-day postoperative AST level (U/L) | 435 (232.75–993.25) |
| EAD | 151 (36.0%) |
| Peak 7-day postoperative Bilirubin level (umol/L) | 105.45 (60.235–174.075) |
| Peak 7-day postoperative Creatinine level (umol/L) | 89.0 (72.0–122.75) |
| Peak 7-day postoperative INR level | 1.47 (1.31–1.7) |
| Biliary complication | 34 (8.1%) |
| Acute rejection | 18 (4.3%) |
| Exceeding Milan criteria[a] | 281 (66.9%) |
| Previous TACE | 151 (36.0%) |

**Table 1** (*continued*)

|  | Number (%) or median (IQR) |
|---|---|
| Previous radiofrequency ablation | 98 (23.3%) |
| Both TACE &radiofrequency ablation | 173 (47.3%) |
| Hepatectomy before LT | 60 (14.3%) |
| AFP (ng/ml) | 29.17 (5.26–541.65) |
| Tumor size (mm) | 53.21 ± 40.23 |
| Poor tumor differentiation | 120 (28.6%) |
| Microscopic vascular invasion | 110 (26.2%) |
| Presence of a satellite nodule | 259 (61.7%) |
| HCC recurrence | 133 (31.7%) |
| Recurrence time (m) | 16 (9–32) |

Notes.

[a] Based on clinical staging.

Abbreviations: DCD, donation after circulatory death; AFP, indicates serum $\alpha$-fetoprotein; TACE, transarterial chemoembolization; MELD, model for end-stage liver disease; EAD, early allograft dysfunction; LT, liver transplant; HCC, hepatocellular carcinoma; PRBC, packed red blood cells; AST, serum aspartate aminotransferase; ALT, serum alanine aminotransferase; INR, international normalized ratio.

**Table 2** Differences in factors represent IRI in clinic between CIT>568 min and CIT ≤568 min.

|  | CIT > 568(n = 72) | CIT ≤568(n = 348) | P |
|---|---|---|---|
| Peak 7-day postoperative ALT level (U/L) | 957 (552.75–1,355.5) | 593 (322.5–957.1) | 0.000 |
| Peak 7-day postoperative AST level (U/L) | 712.5 (381–1,374.25) | 390 (219.5–917.5) | 0.000 |
| Peak 7-day postoperative TBIL level (umol/L) | 134.05 (72.525–209.35) | 99.5 (57.375–167.225) | 0.012 |
| Peak 7-day postoperative Crea level (umol/L) | 90 (74–125) | 89 (71–121.75) | 0.373 |
| Peak 7-day postoperative INR level (S) | 1.525 (1.31–1.75) | 1.46 (1.3125–1.6875) | 0.277 |
| Biliary complication | 4 (5.6%) | 30 (8.6%) | 0.385 |
| Acute rejection | 3 (4.2%) | 15 (4.3%) | 0.956 |
| EAD | 36 (50%) | 115 (33.0%) | 0.006 |
| Postoperative respiratory support (hour) | 20 (12–45) | 16 (11–28) | 0.045 |
| Infusion of FFP (Unit) | 7 (5–12) | 6.5 (4–9) | 0.02 |

Notes.

Abbreviations: AST, serum aspartate aminotransferase; ALT, serum alanine aminotransferase; TBIL, total bilirubin; Crea, creatinine; INR, international normalized ratio; EAD, early allograft dysfunction; FFP, fresh frozen plasma.

the recurrence group (59.4% *vs.* 48.8%, $p = 0.043$), but a lower rate of diabetes among recipients (7.5% *vs.* 15.3%, $p = 0.026$). Recipients in the recurrence group had a higher MELD score (15.287 ± 9.48 *vs.* 14.233 ± 8.36, $p = 0.023$), and significantly higher TBIL levels before LT (32.3 *vs.* 27.4 umol/L, $p = 0.002$). Furthermore, the recurrence group had a significantly higher percentage of patients exceeding the Milan criteria (79.7% *vs.* 61.0%, $p = 0.000$), larger tumor size (66.96 ± 45.24 *vs.* 46.83 ± 36.02 mm, $p = 0.000$), poorer tumor differentiation (46.6% *vs.* 28.6%, $p = 0.006$), and a higher percentage of microscopic vascular invasion (46.6% *vs.* 16.7%, $p = 0.000$). Lastly, the proportion of recipients with preoperative AFP ≥ 400 ng/ml was significantly higher in the recurrence group (42.9% *vs.* 20.2%, $p = 0.000$) (Table 3).

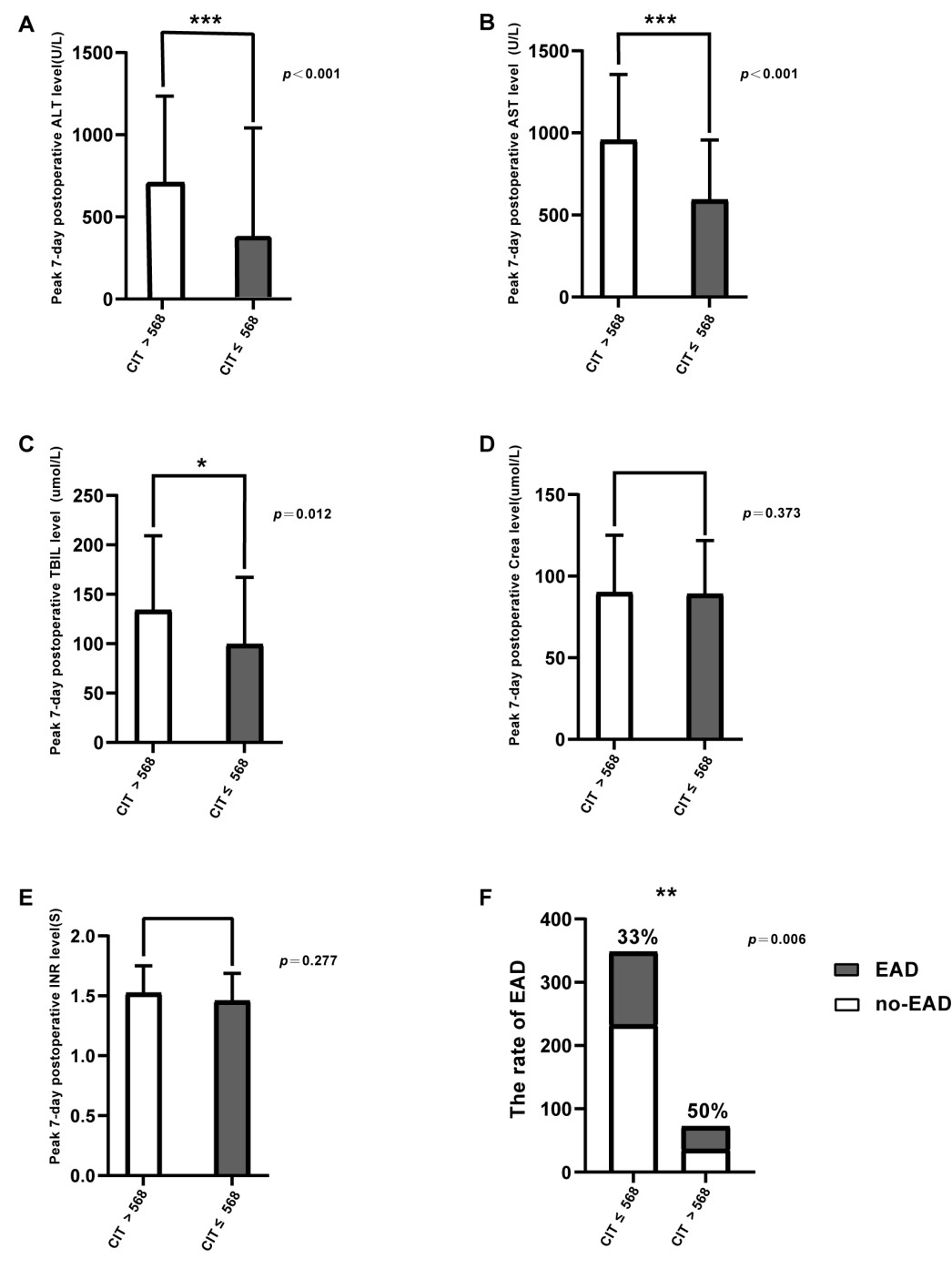

**Figure 1** **Comparison of posttransplant outcomes by CIT.** (A) Peak 7-day postoperative serum alanine aminotransferase (ALT) level (U/L), $p < 0.001$. (B) Peak 7-day postoperativeserum aspartate aminotransferase (AST) level (U/L), $p < 0.001$. (C) Peak 7-day postoperative bilirubin level (umol/L), $p = 0.012$. (D) Peak 7-day postoperative creatinine level (umol/L), $p = 0.373$. (E) Peak 7-day postoperative international normalized ratio (INR) level, $p = 0.277$. (F) The incidence of early allograft dysfunction (EAD), $p = 0.006$. $* \ p < 0.05$, $** \ p < 0.01$, $*** \ p < 0.001$.

**Table 3** Differences in donor factors and recipient factors variables between patients with ($n = 133$) and without ($n = 287$) HCC recurrence.

| | HCC recurrence ($n = 133$) | No HCC recurrence ($n = 287$) | P |
|---|---|---|---|
| Donor Factors | | | |
| Age (years) | 37.05 ± 14.25 | 36.95 ± 14.43 | 0.946 |
| Body mass index | 22.04 ± 2.94 | 23.37 ± 22.04 | 0.492 |
| Gender (male) | 115 (86.5%) | 253 (88.2%) | 0.626 |
| Warm ischemia time (minutes) (DCD donor) | 5 (3.5–7) | 6 (5–10) | 0.291 |
| Cold ischemia time >568 (min) | 35 (26.32%) | 37 (12.89%) | 0.001 |
| Sodium level before harvest (mmol/L) | 151.24 ± 15.44 | 154.94 ± 87.03 | 0.615 |
| ALT level before harvest (U/L) | 39 (26–73) | 52 (27–89.11) | 0.219 |
| AST level before harvest (U/L) | 64 (39–130.5) | 60 (30–106.99) | 0.007 |
| TBIL level before harvest (umol/L) | 22.73 ± 17.67 | 24.07 ± 22.32 | 0.543 |
| DCD | 30 (22.6%) | 49 (17.1%) | 0.181 |
| Recipient Factors | | | |
| Age (years) | 50.82 ± 10.93 | 52.95 ± 9.8 | 0.046 |
| Body mass index (kg/m$^2$) | 23.68 ± 3.27 | 23.07 ± 3.31 | 0.079 |
| Gender (male) | 79 (59.4%) | 140 (48.8%) | 0.043 |
| Complications | | | |
| Hypertension | 12 (9.0%) | 44 (15.3%) | 0.077 |
| Diabetes | 10 (7.5%) | 44 (15.3%) | 0.026 |
| Coronary heart disease | 3 (2.3%) | 6 (2.1%) | 0.913 |
| MELD | 15.287 ± 9.48 | 14.233 ± 8.36 | 0.023 |
| ALT level before LT (U/L) | 39 (26–73) | 37 (22–67) | 0.544 |
| AST level before LT (U/L) | 64 (39–130) | 46 (33–98) | 0.868 |
| TBIL level before LT (umol/L) | 32.3 (16.15–119.6) | 27.4 (17.1–50.9) | 0.002 |
| Operation time (min) | 467.77 ± 114.71 | 456.89 ± 104.31 | 0.687 |
| Intraoperative blood loss (ml) | 1,500 (800–2,300) | 1,200 (800–2,000) | 0.503 |
| Intraoperative PRBC transfusions (unit) | 4 (2–8) | 4 (2–6) | 0.391 |
| anhepatic period (min) | 60.29 ± 45.5 | 55.79 ± 19.58 | 0.157 |
| Infusion of FFP (Unit) | 6.5 (4.5–9.3) | 6.5 (4–9) | 0.399 |
| Peak 7-day postoperative ALT level (U/L) | 723 (336.5–1,142) | 647 (346–976) | 0.304 |
| Peak 7-day postoperative AST level (U/L) | 444 (236–947.5) | 422 (231–1,002.51) | 0.939 |
| Peak 7-day postoperative Tbil level (U/L) | 108 (56.05–180.65) | 104.3 (62.9–170.5) | 0.417 |
| Peak 7-day postoperative Creatinine level (umol/L) | 85 (68–113.5) | 90 (73–125) | 0.117 |
| Peak 7-day postoperative INR level | 1.46 (1.30–1.67) | 1.48 (1.32–1.71) | 0.326 |
| EAD | 46 (34.6%) | 105 (36.6%) | 0.691 |
| Biliary complication | 11 (8.3%) | 23 (8.0%) | 0.928 |
| Acute rejection | 6 (4.5%) | 12 (4.2%) | 0.877 |
| Exceeding Milan criteria | 106 (79.7%) | 175 (61.0%) | 0.000 |

**Table 3** (*continued*)

|  | HCC recurrence (*n* = 133) | No HCC recurrence (*n* = 287) | *P* |
|---|---|---|---|
| Previous TACE | 48 (36.1%) | 103 (35.9%) | 0.832 |
| Previous radiofrequency ablation | 34 (25.6%) | 64 (22.3%) | 0.509 |
| Hepatectomy before LT | 20 (15.0%) | 40 (13.9%) | 0.917 |
| AFP (ng/ml) ≥400 | 57 (42.9%) | 58 (20.2%) | 0.000 |
| Tumor size (mm) | 66.96 ± 45.24 | 46.83 ± 36.02 | 0.000 |
| Poor tumor differentiation | 56 (42.1%) | 82 (28.6%) | 0.006 |
| Microscopic vascular invasion | 62 (46.6%) | 48 (16.7%) | 0.000 |

**Notes.**

Abbreviations: DCD, donation after circulatory death; AFP, indicates serum $\alpha$-fetoprotein; TACE, transarterial chemoembolization; MELD, model for end-stage liver disease; EAD, early allograft dysfunction; LT, liver transplant; HCC, hepatocellular carcinoma; AST, serum aspartate aminotransferase; ALT, serum alanine aminotransferase; TBIL, total bilirubin; PRBC, packed red blood cells; INR, international normalized ratio; FFP, fresh frozen plasma.

**Table 4  Multivariate analysis for risk factors of HCC recurrence.**

|  | OR (95% CI) | *P* value[*] |
|---|---|---|
| Donor characteristics |  |  |
| Cold ischemia time >568 | 2.406 (1.371–4.220) | 0.002 |
| AST level before harvest (U/L) | 1.000 (0.999–1.001) | 0.654 |
| Recipient characteristics |  |  |
| Age (years) | 1.018 (0.996–1.041) | 0.116 |
| Gender (male) | 1.300 (0.668–2.529) | 0.440 |
| Diabetes | 0.706 (0.338–1.473) | 0.353 |
| MELD | 0.994 (0.969–1.019) | 0.615 |
| TBIL level before LT (umol/L) | 1.000 (0.997–1.002) | 0.792 |
| Exceeding Milan criteria | 2.135 (1.269–3.590) | 0.004 |
| AFP (ng/ml) ≥400 | 2.195 (1.360–3.543) | 0.001 |
| Tumor numbers >3 | NA※ | NA※ |
| Tumor size (mm)>50 | NA※ | NA※ |
| Poor tumor differentiation | 1.612 (1.009–2.577) | 0.046 |
| Microscopic vascular invasion | NA※ | NA※ |

**Notes.**

[*]Logistic regression model analysis.

NA※, Tumor size, number of tumor, and MVI were not included regression model, as they are included in Milan criteria.

Abbreviations: MELD, model for end-stage liver disease; HCC, hepatocellular carcinoma; AST, serum aspartate aminotransferase; TBIL, total bilirubin.

Next, the factors that differed between the two groups were analyzed in a multivariate analysis by using logistic regression. Cold ischemia time > 568 min (OR, 2.406; 95% CI [1.371–4.220]; $p = 0.002$), Exceeding Milan criteria (OR, 2.135; 95% CI [1.269–3.590]; $p = 0.004$), AFP (ng/ml) ≥400 (OR, 2.195; 95% CI [1.360–3.543]; $p = 0.001$), poor tumor differentiation (OR, 1.612; 95% CI [1.009–2.577]; $p = 0.046$) were independent predictors of recurrence (Table 4).

**CIT > 568 min is an independent risk factor for DFS but not for OS**

Table 5 lists the parameters related to risk of disease-free survival (DSF). Multivariate Cox regression analysis showed that the recipient gender (male) (OR, 1.774; 95% CI [1.241–2.536]; $p = 0.002$), AFP ≥ 400 (ng/ml) (OR, 2.032; 95% CI [1.415–2.920]; $p = 0.009$), exceeding Milan criteria (OR, 1.782; 95% CI [1.156–2.747]; $p = 0.009$) and poor tumor differentiation (OR, 1.48; 95% CI [1.041–2.116]; $p = 0.029$) were independent risk factors for DSF (Table 6). DFS rates at 1-and 3-year in CIT > 568 min group were 71.9% and 45.7%, respectively; 79.2% and 66.3%, respectively in CIT ≤568 min group ($p = 0.018$, Fig. 2A). Table 6 lists the parameters related to risk of overall survival (OS), and this is significant in univariate analysis. In multivariate Cox regression analysis, intraoperative PRBC transfusions (unit) (OR, 1.046; 95% CI [1.018–1.076]; $p = 0.001$), AFP ≥ 400 (ng/ml) (OR, 2.42; 95% CI [1.648–3.554]; $p < 0.001$) and poor tumor differentiation (OR, 1.48; 95% CI [1.041–2.116]; $p = 0.017$) were independent risk factors for overall survival (Table 5). OS rates at 1-and 5-year in CIT > 568 min group were 86% and 53%, respectively; 87.4% and 54.1%, respectively in CIT ≤568 min group ($p = 0.533$, Fig. 2B).

## DISCUSSION

The quality of donor grafts is one of the most important factors affecting ischemia-reperfusion injury of LT. Moreover, ischemic time, cold ischemic time, and warm ischemic time are vital indices for evaluating the quality of the donor's liver. This study showed that a prolongation of CIT elevated the AST, ALT, and bilirubin levels to 7 days after LT. Unsurprisingly, the EAD incidence was significantly higher in the CIT > 568 min group. This result agrees with the findings of a previous study regarding ischemia-reperfusion and EAD (*Ito et al., 2021*). In addition, more intraoperative infusion of FFP and longer respiratory support time suggested a worse recovery in CIT > 568 min group. Currently, LT is an optional treatment for HCC based on the tumor characteristics. However, research on donor-related factors and their impact on tumor recurrence is still insufficient. Numerous studies have argued that liver ischemia is a risk factor for liver cancer recurrence (*Yoshimoto et al., 2012*; *Yang et al., 2019*; *Kornberg et al., 2015b*).

This study demonstrated that the CIT of liver grafts is an independent risk factor for the tumor recurrence after LT in patients with HCC. Another study of 103 LT patients reported that prolonged mean CIT and WIT promoted the risk of HCC recurrence (*Fisher et al., 2007*). The survival rate of patients and grafts was lower in HCC recipients who received DCD allografts which suggested that IRI injury had a greater impact on HCC recipients, despite the lack of HCC recurrence data (*Kornberg et al., 2015a*). A large retrospective studyused the data of 9,724 patients with HCC who received an LT to conclude that prolonged donor warm ischemia time was a risk factor for HCC recurrence (*Croome et al., 2013*). A single-center study elaborated that AST ≥ 1,896 U/L was a risk factor for HCC recurrence after LT with DBD grafts (*Duvoux et al., 2012*). Moreover, donor age > 60 years and donor WIT were the risk factors for increased HCC recurrence (*Orci et al., 2018*). Thus, these organs are susceptible to intensified IRI, which may lead to higher rates of HCC recurrence after LT.

**Table 5  Prognostic factors for disease-free survival on univariate and multivariate analysis.**

| Variables | Univariate | | Multivariate | |
|---|---|---|---|---|
| | OR (95% CI) | *P* value* | OR (95% CI) | *P* value* |
| Donor characteristics | | | | |
| Age (years) | 1.004 (0.992–1.017) | 0.485 | | |
| Body mass index (kg/m²) | 0.996 (0.974–1.018) | 0.714 | | |
| Gender (male) | 1.247 (0.821–1.893) | 0.3 | | |
| Warm ischemia time (minutes)(80) | 0.944 (0.844–1.057) | 0.318 | | |
| Cold ischemia time >568 | 1.688 (1.146–2.484) | 0.008 | 1.652 (1.111–2.458) | 0.013 |
| Sodium level before harvest (mmol/L) | 0.999 (0.994–1.004) | 0.566 | | |
| ALT level before harvest (U/L) | 1.001 (1.000–1.002) | 0.125 | | |
| AST level before harvest (U/L) | 1.0 (0.999–1.001) | 0.807 | | |
| TBIL level before harvest (umol/L) | 0.998 (0.989–1.007) | 0.651 | | |
| DCD | 0.866 (0.574–1.309) | 0.495 | | |
| Recipient characteristics | | | | |
| Age (years) | 0.986 (0.970–1.003) | 0.098 | | |
| Body mass index (kg/m²) | 1.046 (0.995–1.101) | 0.080 | | |
| Gender (male) | 0.982 (0.588–1.640) | 0.982 | | |
| Complications | | | | |
| Hypertension | 1.627 (0.899–2.945) | 0.108 | | |
| Diabetes | 1.951 (0.991–3.839) | 0.053 | | |
| Coronary heart disease | 1.028 (0.327–3.232) | 0.963 | | |
| MELD | 1.022 (1.003–1.041) | 0.022 | 1.013 (0.995–1.032) | 0.155 |
| ALT level before LT (U/L) | 1.0 (0.999–1.000) | 0.544 | | |
| AST level before LT (U/L) | 1 (1–1) | 0.868 | | |
| TBIL level before LT (umol/L) | 1.002 (1.001–1.003) | 0.052 | | |
| Operation time (min) | 1.000 (0.999–1.002) | 0.885 | | |
| Intraoperative blood loss (ml) | 1 (1–1) | 0.687 | | |
| Intraoperative PRBC transfusions (unit) | 1.013 (0.987–1.042) | 0.391 | | |
| anhepatic period (min) | 1.001 (0.996–1.005) | 0.817 | | |
| Peak 7-day postoperative ALT level (U/L) | 1 (1–1) | 0.256 | | |
| Peak 7-day postoperative AST level (U/L) | 1 (1–1) | 0.205 | | |
| EAD | 1.219 (0.85–1.749) | 0.282 | | |
| Peak 7-day postoperative Creatinine level (umol/L) | 1.0 (0.998–1.002) | 0.923 | | |
| Peak 7-day postoperative INR level | 0.999 (0.997–1.001) | 0.307 | | |
| | 0.872 (0.602–1.259) | 0.872 | | |
| Biliary complication | 1.065 (0.574–1.974) | 0.842 | | |
| Acute rejection | 1.298 (0.572–2.950) | 0.533 | | |
| Infusion of FFP (Unit) | 0.999 (0.993–1.004) | 0.611 | | |
| Exceeding Milan criteria[a] | 2.085 (1.366–3.184) | 0.001 | 1.742 (1.129–2.687) | 0.012 |
| Previous TACE | 1.039 (0.729–1.481) | 0.832 | | |
| Previous radiofrequency ablation | 0.877 (0.593–1.296) | 0.509 | | |

| Variables | Univariate | | Multivariate | |
|---|---|---|---|---|
| | OR (95% CI) | P value[*] | OR (95% CI) | P value[*] |
| Hepatectomy before LT | 1.026 (0.637–1.651) | 0.917 | | |
| AFP (ng/ml) ≥400 | 2.953 (2.093–4.166) | 0.000 | 2.410 (1.679–3.461) | 0.000 |
| Tumor numbers >3 | 1.668 (1.185–2.347) | 0.003 | NA※ | NA※ |
| Tumor size (mm)>50 | 1.518 (1.078–2.136) | 0.017 | NA※ | NA※ |
| Poor tumor differentiation | 1.749 (1.236–2.475) | 0.002 | 1.547 (1.086–2.203) | 0.016 |
| Microscopic vascular invasion | 3.564 (2.524–5.033) | 0.000 | NA※ | NA※ |

**Notes.**

*Cox's proportional regression analysis.

NA※, Tumor size, number of tumor, and MVI were not included to avoid colinearity, as they are included in Milan criteria.

Abbreviations: DCD, donation after circulatory death; AFP, indicates serum $\alpha$-fetoprotein; TACE, transarterial chemoembolization; MELD, model for end-stage liver disease; EAD, early allograft dysfunction; LT, liver transplant; HCC, hepatocellular carcinoma; PRBC, packed red blood cells; AST, serum aspartate aminotransferase; ALT, serum alanine aminotransferase; TBIL, total bilirubin; INR, international normalized ratio; FFP, fresh frozen plasma.

In our study, 30 of 80 patients, who received a liver graft from DCD donors, experienced HCC recurrence. Compared with the DBD group, the HCC recurrence rate was not statistically significant ($p = 0.642$; univariate Cox's regression analysis, the reference to the data file as a Supplementary File), which might be due to the small sample size.

Although numerous studies have reported that IRI could elevate the risk of HCC recurrence (*Grat et al., 2018*; *Orci et al., 2018*; *Croome et al., 2015*), the mechanisms are not yet fully understood owing to the complex network of cellular and molecular interactions. Moreover, the mechanism of injury caused by different forms of IRI varies, and the alteration of the hepatic microenvironment could create a "fertile soil" for tumor cells. For instance, sinusoidal endothelial cells (SECs) are vulnerable to the IRI, particularly during cold ischemia, which may be the risk factor for endothelial cell swelling, unbalanced vasoconstriction, and neutrophil plugging (*Tejima et al., 2004*). In addition, IRI can break the barrier between hepatocytes and the blood, making molecules and cells spread easily. Microcirculatory disturbances lead to tissue ischemia and hypoxia. It also activates the HIF-1 $\alpha$ pathway, which has been recognized as a vascular endothelial growth factor associated with metastatic disease (*Semenza, 2012*). IRI-induced inflammatory cytokines, tumor necrosis factor- $\alpha$, or interleukin-1 may promote the metastasis of different cancer cells through the expression of adhesion molecules (such as E-selectin, ICAM-1, and VCM-1), acting as mediators of tumor growth (*Yoshimoto et al., 2012*; *Reina & Espel, 2017*; *Kong et al., 2018*).

Damage-associated molecular patterns (DAMPs) are other important mechanisms of IRI, promoting tumor progression, which has been released by injured hepatic cells and may perpetuate the non-infectious sterile inflammatory response. The DAMPs released into the blood can cause innate immune activation and robust hepatocellular injury. For instant, high-mobility group protein B1 (HMGB1), which is the most frequently encountered DAMP in cells undergoing stress, could translocate from the nucleus to the cytosol, bind to mitochondrial DNA (mtDNA), and be released by damaged mitochondria during hypoxia (*Srikrishna & Freeze, 2009*). The complex of HMGB1 and mtDNA subsequently activates the TLR9 signaling pathways to promote tumor cell proliferation (*Liu et al., 2015*). *Tohme*

**Table 6  Prognostic factors for overall survival on univariate and multivariate analysis.**

| Variables | Univariate | | Multivariate | |
|---|---|---|---|---|
| | OR (95% CI) | *P* value[*] | OR (95% CI) | *P* value[*] |
| Donor characteristics | | | | |
| Age (years) | 1.01(0.997–1.023) | 0.137 | | |
| Body mass index (kg/m$^2$) | 1 (0.985–1.015) | 0.986 | | |
| Gender (male) | 1.069 (0.701–1.629) | 0.757 | | |
| Warm ischemia time (minutes)(80) | 0.913 (0.823–1.012) | 0.084 | | |
| Cold ischemia time >568 | 1.143 (0.751–1.738) | 0.533 | | |
| Sodium level before harvest (mmol/L) | 0.997 (0.989–1.006) | 0.559 | | |
| ALT level before harvest (U/L) | 1.001 (1.000–1.002) | 0.046 | NA# | NA# |
| AST level before harvest (U/L) | 1.0 (1.0–1.001) | 0.191 | | |
| TBIL level before harvest (umol/L) | 1.005 (0.998–1.013) | 0.143 | | |
| DCD | 0.866 (0.574–1.309) | 0.495 | | |
| Recipient characteristics | | | | |
| Age (years) | 1.002 (0.985–1.020) | 0.789 | | |
| Body mass index (kg/m$^2$) | 1.021 (0.966–1.078) | 0.467 | | |
| Gender (male) | 0.962 (0.585–1.581) | 0.877 | | |
| Complications | | | | |
| Hypertension | 1.315 (0.805–2.148) | 0.273 | | |
| Diabetes | 1.187 (0.654–2.155) | 0.572 | | |
| Coronary heart disease | 1.316 (0.325–5.336) | 0.7 | | |
| MELD | 1.028 (1.009–1.047) | 0.004 | 1.016 (0.996–1.036) | 0.114 |
| ALT level before LT (U/L) | 1.0 | 0.510 | | |
| AST level before LT (U/L) | 1 | 0.858 | | |
| TBIL level before LT (umol/L) | 1.002 (1.001–1.003) | 0.000 | | |
| Operation time (min) | 1.001 (0.999–1.002) | 0.45 | | |
| Intraoperative blood loss (ml) | 1 (1–1) | 0.084 | | |
| Intraoperative PRBC transfusions (unit) | 1.036 (1.010–1.063) | 0.007 | 1.048 (1.019–1.078) | 0.001 |
| anhepatic period (min) | 1.00 (0.996–1.005) | 0.863 | | |
| Peak 7-day postoperative ALT level (U/L) | 1 (1–1) | 0.000 | NA# | NA# |
| Peak 7-day postoperative AST level (U/L) | 1 (1–1) | 0.034 | NA# | NA# |
| EAD | 1.172 (0.819–1.677) | 0.385 | | |
| Peak 7-day postoperative Bilirubin level (umol/L) | 1.001 (1.000–1.003) | 0.104 | | |
| Peak 7-day postoperative Creatinine level (umol/L) | 0.999 (0.997–1.002) | 0.610 | | |
| Peak 7-day postoperative INR level | 0.975 (0.857–1.110) | 0.706 | | |
| Biliary complication | 1.045 (0.547–1.996) | 0.895 | | |
| Acute rejection | 1.230 (0.501–3.018) | 0.651 | | |
| Infusion of FFP (Unit) | 1 (0.999–1.001) | 0.938 | | |
| Exceeding Milan criteria[a] | 1.8 (1.174–2.759) | 0.007 | 1.388(0.895–2.154) | 0.143 |
| Previous TACE | 1.419 (0.962–2.092) | 0.078 | | |
| Previous radiofrequency ablation | 1.241 (0.794–1.940) | 0.344 | | |
| Hepatectomy before LT | 1.25 (0.727–2.147) | 0.42 | | |

| Variables | Univariate | | Multivariate | |
|---|---|---|---|---|
| | OR (95% CI) | *P*value[*] | OR (95% CI) | *P*value[*] |
| AFP (ng/ml) ≥400 | 3.195 (2.239–4.558) | 0.000 | 2.915 (1.982–4.287) | 0.000 |
| Tumor numbers >3 | 1.819 (1.276–2.594) | 0.001 | NA※ | NA※ |
| Tumor size (mm)>50 | 1.557 (1.093–2.218) | 0.014 | NA※ | NA※ |
| Poor tumor differentiation | 1.921 (1.343–2.746) | 0.000 | 1.577 (1.083–2.298) | 0.018 |
| Microscopic vascular invasion | 3.038 (2.115–4.362) | 0.000 | NA※ | NA※ |

**Notes.**

[*]Cox's proportional regression analysis.

[NA#]not applicable.

NA※, Tumor size, number of tumor, and MVI were not included to avoid colinearity, as they are included in Milan criteria.

Abbreviations: DCD, donation after circulatory death; AFP, indicates serum $\alpha$-fetoprotein; TACE, transarterial chemoembolization; MELD, model for end-stage liver disease; EAD, early allograft dysfunction; LT, liver transplant; HCC, hepatocellular carcinoma; PRBC, packed red blood cells; AST, serum aspartate aminotransferase; ALT, serum alanine aminotransferase; TBIL, total bilirubin; INR, international normalized ratio; FFP, fresh frozen plasma.

*et al. (2017)* found that upregulated HMGB1 could enhance mitochondrial biogenesis in HCC cancer cells and promote tumor survival and proliferation.

Many marginal donor livers are used clinically owing to the lack of a donation pool. This issue can lead to more serious IRI damage, subsequent complications, and HCC recurrence after transplantation. Thus, evaluating liver function and repairing the liver *in vitro* is vital to reducing complications after transplantation. Traditional static cold storage (SCS) may be impossible. With the progress of perfusion technology and the upgrading of perfusion machines, mechanical perfusion has been applied significantly for marginal liver donation. The process is effective in refurnishing energy stores during preservation to resist impending ischemia-reperfusion injury after organ implant. In addition, *ex situ* organ perfusion simulates physiological blood reflow, reflects the quality of organs, and provides an objective functional evaluation for transplant surgeons. Currently, hypothermic (hypothermic oxygenated perfusion, HOPE) and normothermic (normothermic machine perfusion, NMP) and their combination are widely used machine perfusion approaches (*Karangwa et al., 2016*). Several clinical trials have confirmed that the HOPE technique offers promising advantages compared with SCS in patients whose organs are transplanted from DCD donors (*Karangwa et al., 2016*; *Schlegel et al., 2019*). Another group reported an encouraging result, revealing the incidence of symptomatic non-anastomotic biliary strictures in dual hypothermic oxygenated machine perfusion (D-HOPE), which was perfused through the hepatic artery and portal vein, showing that liver graft volumes in 78 patients were significantly lower after LT than in the SCS group (78 patients) after LT from DCD donors (*Schlegel et al., 2013*). Ischemia-free organ transplantation (IFOT) eliminates the ischemia and hypoxia of the organ by uninterrupted NMP during the procurement and implant periods. A retrospective study also demonstrated that patients who underwent the IFOT had significant higher DFS rates at 1 and 3 years than the SCS group after LT in recipients with HCC (*Tang et al., 2021*).

The disadvantages of ischemia-reperfusion in treating hepatocellular carcinoma have been documented. Compared with hepatectomy, LT is more vulnerable to IRI because of the sophisticated segments and procedures. However, the assessment of the patients' oncological risk profile following LT is limited to the clinical characteristics and biological

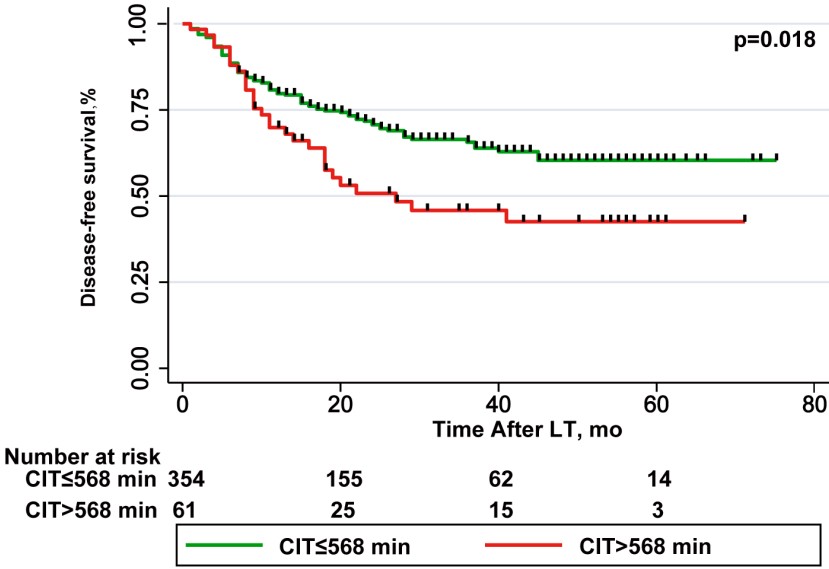

**A.Disease-free survival**

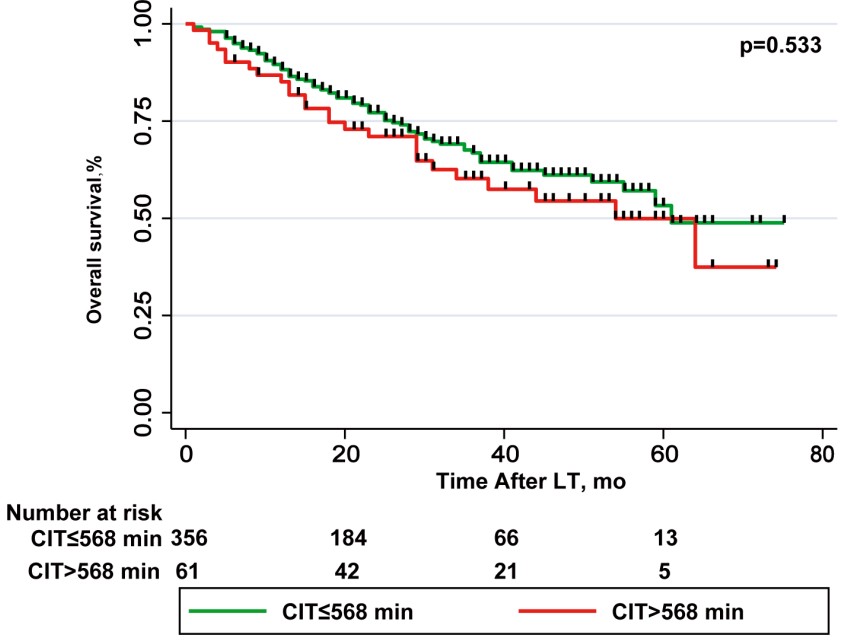

**B.Overall survival**

**Figure 2 Comparison of disease-free survival and overall survival in CIT ≤568 min group and CIT >568 min group.** (A) Disease-free survival (DSF) rate was significantly higher after LT with CIT ≤ 568 min as compared to LT with CIT >568 min, $p = 0.018$. (B) Overall survival (OS) rate was lower after LT with CIT >568 min as compared to LT with CIT ≤ 568 min, difference was not statistically significant, $p = 0.533$.

features of carcinoma. CIT is the easiest to evaluate compared with complicated biological and histological tests; although CIT is not the most direct evidence of IRI. Our results show that prolonged CIT can significantly increase the peak level of ALT, AST, TBIL, and the incidence of EAD in patients after LT.

Although several studies had found that prolonged cold ischemia time promotes recurrence after LT for HCC, their research did not cover East Asian people. Our study retrospectively analyzed 420 Chinese HCC patients with LT, demonstrated that CIT > 568 min was an independent risk factor for recurrence after LT in patients with HCC, which move forward the CIT window that warns of HCC recurrence. Therefore, the significance of CIT in patients undergoing LT for HCC should be re-evaluated.

Our study is a single-center retrospective study, which is its limitation. Future studies must conduct multicenter, prospective cohort studies and explore the biological mechanism behind IRI and tumor recurrence, which may guide the selection of grafts for HCC patients undergoing LT, and provide therapeutic targets for reducing ischemia-reperfusion and recurrence after LT.

## CONCLUSION

In conclusion, our study confirms the harmful effect of cold ischemia time on the prognosis of liver transplantation patients with HCC. Cold ischemia time should be reconsidered when evaluating patients with HCC on the waiting list. Shortening the cold ischemia time should be one of the effective strategies to improve the prognosis, especially for those with relatively low tumor load and long expected survival.

### Funding
This work was supported by the Guangdong Provincial Funds for High-end Medical Equipment (No. 2020B1111140003); the Guangdong Provincial Key Laboratory of Organ Donation and Transplant Immunology, The First Affiliated Hospital, Sun Yat-sen University, Guangzhou, China (No. 2017B030314018, No. 2020B1212060026); the Guangdong Provincial International Cooperation Base of Science and Technology (Organ Transplantation) and The First Affiliated Hospital, Sun Yat-sen University, Guangzhou, China (No. 2020A0505020003). The funders had no role in study design, data collection and analysis, decision to publish, or preparation of the manuscript.

### Grant Disclosures
The following grant information was disclosed by the authors:
Guangdong Provincial Funds for High-end Medical Equipment: 2020B1111140003.
Guangdong Provincial Key Laboratory of Organ Donation and Transplant Immunology, The First Affiliated Hospital, Sun Yat-sen University, Guangzhou, China: 2017B030314018, 2020B1212060026.

Guangdong Provincial International Cooperation Base of Science and Technology (Organ Transplantation), The First Affiliated Hospital, Sun Yat-sen University, Guangzhou, China: 2020A0505020003.

## Competing Interests

The authors declare there are no competing interests.

## Author Contributions

- Jia Yu conceived and designed the experiments, performed the experiments, analyzed the data, prepared figures and/or tables, authored or reviewed drafts of the article, and approved the final draft.
- Tang Yunhua conceived and designed the experiments, performed the experiments, prepared figures and/or tables, and approved the final draft.
- Yiwen Guo performed the experiments, prepared figures and/or tables, and approved the final draft.
- Yuqi Dong performed the experiments, prepared figures and/or tables, and approved the final draft.
- Jin long Gong performed the experiments, authored or reviewed drafts of the article, and approved the final draft.
- Tielong Wang performed the experiments, analyzed the data, authored or reviewed drafts of the article, and approved the final draft.
- Zhitao Chen analyzed the data, prepared figures and/or tables, and approved the final draft.
- Maogen Chen analyzed the data, authored or reviewed drafts of the article, and approved the final draft.
- Weiqiang Ju analyzed the data, authored or reviewed drafts of the article, and approved the final draft.
- Xiaoshun He conceived and designed the experiments, authored or reviewed drafts of the article, and approved the final draft.

## Data Availability

The raw data is available in the Supplementary File.

## Supplemental Information

Supplemental information for this article can be found online at http://dx.doi.org/10.7717/peerj.18126#supplemental-information.

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
