# Peer review of "Beyond graft function impairment after liver transplantation: the prolonged cold ischemia time impact on recurrence of hepatocellular carcinoma after liver transplantation—a single-center retrospective study"

_PeerJ, doi:10.7717/peerj.18126_

## Round 0.1 · original submission · Major Revisions

The authors are requested to carefully revise the manuscript and answer the questions raised by the reviewers.

Reviewer 1 ·

Basic reporting

This is an interesting study on the oncologic impact of cold ischemia time in liver transplant patients with HCC. The manuscript is well written, the data confirms previous studyies on the deleterious impact CIT and ischemia/reperfusion injury folling LT for liver cancer. The manuscript is worth being published. However, discussion section is too long. Especially the part describing potential cancerogenic mechanisms of ischemi time should be significantly shortended. The authors shoud comment on the fact that CIT was identified as an independent prognostic factor regarding HCC recurrence but not overall survival.

Experimental design

Experimental design is appropriate

Validity of the findings

Meeting standard criteria; discussion has to be shortended.

Additional comments

no additional comments

·

Basic reporting

Dear Sirs,
Thank you for the opportunity to review.
English is inaccurate, with several flaws and inappropriate terms used throughout. The paper requires linguistic revision.
The study methodology is also inaccurate. The authors should've implemented a univariable and multivariable analysis to identify risk factors for HCC recurrence and later a hierarchical analysis to define the role of CIT over other variables and parameters.
I suggest the authors would seek help from an experienced statistician.

Experimental design

The study design is inaccurate as described above.

Validity of the findings

Doubtful.

Additional comments

The paper is difficult to follow.

Reviewer 3 ·

Basic reporting

-I suggest thorough revision of grammar. There are many redundancies and grammar problems.

-L 115 Mean or median?

-L 115: “Most donors were from … “ This sentence is hard to understand. What does the 7.1 % refer to?

-L 119: “The mean total CIT was 443.6 min, and the median WIT was 5 min” Why change statistics?

-L 139 “The biliary complication incidence was 5.6% in the CIT > 568 min group, and 8.6% in CIT f 568 min group”. Was this difference significant?

-L 150-l 154: Why are the results from survival in between paragraphs of univariate analysis on HCC recurrence?
-L 160: I would suggest to combine the paragraph of survial analysis and cox-regression. The manuscript is quite hard to read in its current structure.

-When did recurrence occur and had time of recurrence impact on survival? (Early vs late recurrence?)

-L 157: Was within MILAN a risk factor or protective factor for HCC recurrence?

-What was median follow-up?

Experimental design

-L 122: More than two-third in the cohort exceeded MILAN. Why?

-L 159: How was “poor tumor differentiation” defined? G2 or G3?

Validity of the findings

-As far as I understand, in multivariate analysis, CIT did not reach significance? This should be discussed. With the high percentage of patients outside MILAN, there might be more impactful variables on recurrence than CIT.
Also, CIT did not have impact on overall survival.

-66% were outside MILAN and 30% (literature: around 20%) recurrence rate was recorded. This shoud be discussed.

Additional comments

-

---

## Round 0.2 · accepted · Accept

After revisions, two reviewers agreed to publish the manuscript. I also reviewed the manuscript and found no obvious risks to publication. Therefore, I also approved the publication of this manuscript.

Reviewer 1 ·

Basic reporting

The authors have appropriately refered to the reviewers criticisms and revised the manuscrpit.

Experimental design

see above

Validity of the findings

see above

Additional comments

see above

·

Basic reporting

Dear Sirs,
I am grateful for your time and effort in amending your paper.
Based on previous comments, the paper has been improved considerably.
Although the findings are nothing new, since the impact of CIT on HCC recurrence has already been documented in the literature, the paper has been restructured to comply with the basic standards of scientific reporting.

Experimental design

Previous inaccuracies have been addressed and resolved.

Validity of the findings

Data look valid, but their novelty is limited.

Additional comments

Please, avoid using the word harvest and replace with procurement.